# Tumor Cells and Cancer-Associated Fibroblasts: An Updated Metabolic Perspective

**DOI:** 10.3390/cancers13030399

**Published:** 2021-01-22

**Authors:** Géraldine Gentric, Fatima Mechta-Grigoriou

**Affiliations:** 1Institut Curie, Stress and Cancer Laboratory, Equipe Labélisée par la Ligue Nationale Contre le Cancer, PSL Research University, 26, rue d’Ulm, F-75248 Paris, France; geraldine.gentric@curie.fr; 2Inserm, U830, 26, rue d’Ulm, F-75005 Paris, France

**Keywords:** metabolism, OXPHOS, cancer cell, CAF, fibroblast, tumor microenvironment, heterogeneity, crosstalk

## Abstract

**Simple Summary:**

Tumors are a complex ecosystem including not only cancer cells, but also many distinct cell types of the tumor micro-environment. While the Warburg effect assessing high glucose uptake in tumors was recognized a long time ago, metabolic heterogeneity within tumors has only recently been demonstrated. Indeed, several recent studies have highlighted other sources of carbon than glucose, including amino acids, fatty acids and lactate. These newly identified metabolic trajectories modulate key cancer cell features, such as invasion capacities. In addition, cancer metabolic heterogeneity is not restricted to cancer cells. Here, we also describe heterogeneity of Cancer-Associated Fibroblast (CAF) subpopulations and their complex metabolic crosstalk with cancer cells.

**Abstract:**

During the past decades, metabolism and redox imbalance have gained considerable attention in the cancer field. In addition to the well-known Warburg effect occurring in tumor cells, numerous other metabolic deregulations have now been reported. Indeed, metabolic reprograming in cancer is much more heterogeneous than initially thought. In particular, a high diversity of carbon sources used by tumor cells has now been shown to contribute to this metabolic heterogeneity in cancer. Moreover, the molecular mechanisms newly highlighted are multiple and shed light on novel actors. Furthermore, the impact of this metabolic heterogeneity on tumor microenvironment has also been an intense subject of research recently. Here, we will describe the new metabolic pathways newly uncovered in tumor cells. We will also have a particular focus on Cancer-Associated Fibroblasts (CAF), whose identity, function and metabolism have been recently under profound investigation. In that sense, we will discuss about the metabolic crosstalk between tumor cells and CAF.

## 1. Introduction

The cancer metabolism field emerged a century ago and is still an intense area of research. This cancer area started with O. Warburg, who discovered that cancer cells consume high levels of glucose and produce ten time more lactic acid than normal cells [1], a process called the “Warburg effect”. In many cancer types, increased glucose uptake as a source of ATP and enhanced glycolytic rates represent a growth advantage for tumor cells. Indeed, aerobic glycolysis represents an effective shift, providing carbon sources to key metabolic pathways required for nucleotide, lipid, amino acid synthesis and antioxidant power. Although the well-known “hallmarks of cancer” described by Hannan and Weinberg have been updated with metabolism reprogramming [2], metabolism research has continued to progress. Importantly, while scientists initially compared the metabolism of cancer cells with their normal counterparts, they have recently begun to compare tumors of the same type with each other. One of the most important discoveries of the last few years breaks with Warburg’s dogma, by demonstrating that carbon from glucose can be used through mitochondrial oxidative phosphorylation (OXPHOS) in a mouse model of glioblastoma [3]. Since this report, several studies have also highlighted metabolic heterogeneity within tumors from the same subtype [4,5,6]. It turns out that glucose is no longer the only source of carbon used by tumors, but that many other sources can be used, such as glutamine, serine, alanine, fatty acids and lactate [6,7,8,9].

Cancer metabolism is no longer restricted to cancer cells. It is now well-established that proliferative and invasive capacities of tumor cells are strongly influenced by their surrounding microenvironment, composed of different cell types, including immune cells, cancer-associated fibroblast (CAF), blood vessels, adipocytes, etc. The tumor ecosystem cooperates to allocate constant nutrient supplies required for tumor growth [10]. In this review, we will discuss how these metabolic interactions occur by considering two major partners: cancer cells and CAF. We will first describe the cancer cell metabolic heterogeneity recently highlighted in several cancer types, and the mechanisms involved in this heterogeneity. We will next describe the different CAF subpopulations recently identified in several cancers and their different energy metabolism programs. Finally, we will discuss the reciprocal metabolic interactions between cancer cells and CAF.

## 2. Cancer Cell Metabolic Heterogeneity

Metabolism is known to generate substrates for anabolic reactions, regulation of signaling pathways, and reduction-oxidation (redox) balance. This raises the possibility that carbon use by cancer cells may affect metabolite accessibility to other cell types. Advances in the cancer field over the last decade have shown how metabolic reprogramming occurs in cancer, and the mechanisms driven metabolic heterogeneity. Here, we will discuss recent studies in the field describing the diversity of carbon sources that can be used or hijacked by tumor cells and the mechanisms involved.

### 2.1. Metabolic Heterogeneity and Carbon-Source Preferences

Tumors require substrates to produce ATP, generate biomass and maintain redox balance (Figure 1). This supplies the high demand for catabolic reactions observed during tumor growth and progression. For decades, tumor cells have been described to predominantly use glucose to generate ATP through glycolysis with lactate production, widely known as the Warburg effect [1]. Still, recently, tumor cells from several cancer types have also been shown to acquire energy through the tricarboxylic acid (TCA) cycle and oxidative phosphorylation (OXPHOS) [6,8,11]. In line with these observations, a new way of understanding metabolic reprogramming in the cancer field has emerged: the metabolism of tumor cells is no more only compared with their normal counterparts, but also between tumors from the same cancer type [4,5,12]. It is becoming increasingly recognized that tumor cells not only capture glucose, but also waste products, such as biosynthetic building blocks. For instance, lactate, previously considered to be the end product of the Warburg effect, contributes to fuel mitochondrial metabolism [13,14]. Similarly, ammonia (NH3) can accumulate in the tumor microenvironment and its assimilation and use by breast cancer cells favor tumor proliferation [15]. In addition, the development of new technologies helped in revealing metabolic heterogeneity and adaptability. We will address these issues by focusing on the most recent publications, previous studies being summarized in earlier reviews [6,7,8,9].

#### 2.1.1. Amino Acids

Although alterations in glucose metabolism are well known to be central in metabolic transformation, several studies have emphasized the role of amino acids in tumor development (see [16,17] for excellent reviews on amino acid contribution in cancer growth and tumor microenvironment). As an example, aspartate has been shown to be important in proliferating cancer cells to enable nucleotide synthesis. Interestingly, one of the most crucial metabolic function achieved by mitochondrial respiration is to support aspartate biosynthesis [18,19,20].

##### Glutamine

Numerous studies have demonstrated that glutamine, a non-essential amino acid, is a major respiratory fuel for tumor cells [21]. A growing amount of evidence shows that glutamine plays a key role in protein, nucleotide and lipid synthesis. Glutamine supplies cellular ATP by replenishing the TCA cycle, a process called anaplerosis. Indeed, glutamine generates α-ketoglutarate (α-KG) and subsequently oxaloacetate to fuel the TCA cycle (Figure 1). In hypoxia conditions, or when mitochondrial function is impaired, an alternative pathway is induced to produce citrate by reductively carboxylating α-KG via NADPH-dependent isocitrate dehydrogenase (IDH) [22,23]. Moreover, glutamine is also a source of glutathione, an important regulator of redox reactions. Some specific cancer types exhibit glutamine dependency [22,24,25,26,27,28,29,30,31,32,33], although they exhibit different mutational profiles, while lung tumors preferentially oxidized glucose instead of glutamine to fuel the TCA cycle [34]. Glutamine addiction has been demonstrated by reports showing that glutamine deprivation inhibits tumor growth [31,32,35]. Moreover, in solid tumors, glutamine level is heterogeneous, being low in the tumor core compared to the periphery [36,37]. Recently, by combining large-scale proteomic analyses and functional assays on high grade ovarian carcinoma (HGSOC), two subgroups of tumors with distinct metabolic profiles have been identified [38]. Although both HGSOC subgroups oxidize glucose to produce lactate, one of the two subgroups consume more preferentially glutamine, through TCA cycle and oxidative phosphorylation (OXPHOS). This subgroup of HGSOC was subsequently referred to as high-OXPHOS, by opposition to the other called low-OXPHOS [38]. Mechanistically, high-OXPHOS HGSOC exhibit a higher oxidative stress compared to low-OXPHOS tumors. This chronic oxidative stress in high-OXPHOS HGSOC activates the promyelocytic leukemia (PML) factor, thereby increasing transcription of electron transport chain (ETC) complexes through activation of peroxisome proliferator-activated receptor gamma coactivator 1-alpha (PGC-1α) [38]. Another recent study highlighted the crosstalk between chronic oxidative stress and glutamine use by TCA cycle in cancer [39]. Indeed, 20% of the *KRAS*-mutant lung adenocarcinoma carry loss-of-function mutations in kelch-like ECH associated protein 1 (*KEAP1*) gene [40]. KEAP1 is a negative regulator of nuclear factor erythroid 2-like (NFE2L2, or NRF2), the key transcriptional regulator of antioxidant response [41,42]. Interestingly, through a combination of CRISPR-Cas9-based genetic screens and metabolomic analyses, Nrf2-mutant cells were shown to be dependent on glutamine oxidation through the TCA cycle for proliferation and survival [39]. These different studies reveal a strong link between oxidative stress and glutamine addiction to sustain TCA cycle and ATP production by mitochondria.

##### Alanine

Alanine, a non-essential amino acid, has recently been shown to play an essential role in cancer cell metabolism. Alanine synthesis from pyruvate is mainly located in mitochondrial matrix [43]. Alanine can be produced through glutamine use in cancer cells. Indeed, glutamine can be converted into glutamate by glutaminase. Subsequently, glutamate can be converted into α-KG and amino acids, such as alanine (Figure 1). In the past, alanine has been used as a metabolic bio-tracer in mouse models of prostate cancer [44]. Alanine was recently shown to be a key metabolic compound in pancreatic ductal adenocarcinomas (PDAC) [45]. PDAC are very aggressive cancers with very few effective therapies. Several features influence PDAC aggressiveness, including genomic complexity, hypo-vascularization, dense stromal reaction (desmoplasia)—discussed in the second part of this review and metabolic reprogramming [46]. In *KRAS*-driven tumors, different processes of recycling, such as autophagy, provide nutrients that can fuel TCA cycle, and promote tumor growth [47]. Moreover, Kras-transformed cells are able to consume proteins to recover the degraded amino acids to fulfil TCA cycle [48]. Glutamine metabolism was first identified to be required for PDAC growth through an alternative pathway in which glutamine-derived aspartate is converted into oxaloacetate by aspartate transaminase (GOT1) [28]. Interestingly, by using conditioned medium from pancreatic stellate cells, PDAC cancer cells show minimal changes in glycolysis rate but a significant increase in mitochondrial respiration [45]. Following a deep metabolomic study, alanine—and not glutamine—was identified as the metabolite responsible for the increased respiration detected in PDAC tumor cells, effect mediated through transamination [45]. Indeed, tracing analyses revealed that alanine is significantly incorporated into citrate and isocitrate for fatty acid biosynthesis thus representing a major carbon source for TCA cycle in PDAC [45]. Finally, recent findings suggest that alanine aminotransferase activity is important for stimulating ECM formation through production of αKG in metastatic breast cancer cells [49]. These data shed a new insight into the role of alanine instead of glutamine in tumor metabolism reprogramming.

##### Serine

Recent studies have highlighted the role of serine and glycine in tumor growth [50,51]. Whether and how increased serine synthesis promotes tumor growth is still under intense investigation. Serine and glycine, which can be either imported into cells or de novo synthetized from a derived branching of glycolysis (Figure 1), contribute not only to protein synthesis, but also to glutathione, nucleotide and phospholipid productions. Amplification of the phosphoglycerate dehydrogenase (PHGDH) gene, encoding the first and limiting enzyme involved in de novo serine synthesis, was detected both in breast cancer and melanoma [52,53]. As cancer cells rapidly use exogenous serine, K.H. Vousden’s laboratory showed that subsequent serine deprivation promotes activation of the serine synthesis pathway and suppresses aerobic glycolysis, resulting in an increased flux into the TCA cycle [54]. ^13^C-labeled glucose tracing in human melanoma cells with high metastatic capacity showed an increased contribution of glucose carbons to the serine/glycine pathway, suggesting an enhanced flux through the folate pathway (Figure 1) [55]. Serine plays a key role in feeding one-carbon units to the tetrahydrofolate (THF) cycle and supports both nucleotide synthesis and nicotinamide adenine dinucleotide phosphate (NADPH) production. Interestingly, one-carbon metabolism contributes to the biosynthesis and recycling of functional metabolites, such as ATP, S-adenosyl-methionine (SAM), and NAD(P)H, with important downstream consequences for cancer cell survival. By analyzing the transfer of ^13^C-labeled carbon from methionine and serine into DNA or RNA in colorectal cancer cells, serine was shown to support the methionine cycle through de novo ATP synthesis by triggering the conversion of methionine to SAM [56]. Finally, a recent work from Vander Heiden’s lab demonstrated that increased PHGDH expression promotes tumor progression in mouse models of both melanoma and breast cancer in serine-limited tumor micro-environment [57]. Indeed, PHGDH expression has no impact on the progression of breast cancer implanted in pancreas—a serine-rich environment-, while tumor growth increases when cells are injected in mammary fat pads—a serine-low environment [57]. Moreover, some human breast cancers show a significant association between poor prognosis and expression of genes involved in mitochondrial serine use and one carbon pathway, especially the serine hydroxymethyltransferase (SHMT2) [58]. In conclusion, non-essential amino acids, such as serine, are involved in several anabolic processes that support cancer cell proliferation. Although some cancer cells rely on de novo serine synthesis, others are dependent on exogenous serine for tumor growth. In that case, dietary restriction in serine can reduce tumor growth in some mouse models, but others are less sensitive to serine depletion, consistent with their capacity to upregulate de novo serine synthesis [59].

#### 2.1.2. Fatty Acids

Fatty acids have gained significant interest in cancer metabolism based on their multiple roles as structural components of membranes, secondary messengers in signaling pathways and fuel sources for energy production [60,61]. Fatty acids can be directly incorporated into cells through specialized transporters or de novo synthesized, process normally restricted to hepatocytes and adipocytes, but reactivated in some cancer cells. Fatty acids can also arise from adipose tissue lipolysis or breakdown of triglycerides. Fatty acids have been shown to be the dominant metabolic substrate in prostate cancer [62]. Some studies started to open a new avenue in the cancer metabolism field by comparing tumors with each other and no longer with their normal counterparts [6]. In line with this new analysis, N. Danial’s laboratory recently reported that a subset of diffuse large B lymphoma preferentially oxidized palmitate through TCA cycle (Figure 1), and not glucose or glutamine [4]. Interestingly, they identified an enhanced antioxidant response in association with the increased fatty acid-driven OXPHOS metabolism [4]. In ovarian cancer, the high-OXPHOS subgroup of HGSOC also exhibits an enhanced oxidative stress and uses fatty acids to fuel TCA cycle [38]. The most well-characterized fatty acid transporter is CD36. Interestingly, high CD36 expression has been associated with poor prognosis across several cancer types, such as breast, ovarian and prostate [61]. CD36 overexpression is induced by co-culture of adipocytes and ovarian cancer cell lines and promotes fatty acid uptake [63]. Still, in this study, the authors did not observe any increase in mitochondrial respiration upon co-culture [63], suggesting that CD36-mediated fatty acid uptake is not used for energy production. Interestingly, CD36 also promotes fatty acid uptake, storage and modulates lipid composition in aggressive Pten mutant mouse models of prostate cancer [64]. Moreover, blocking CD36 in patient-derived prostate cancer xenograft (PDX) models reduces tumor growth in CD36^high^ PDX models, suggesting a potential therapeutic benefit of blocking fatty acid uptake in CD36^high^ prostate cancer [64]. Finally, CD36 has been reported to be highly expressed at the surface of a subpopulation of metastasis-initiating cells in oral carcinoma [65]. Importantly, CD36 overexpression in cell lines with low metastatic capacity greatly increases their potential to metastasize in lymph nodes [65]. Reciprocally, CD36 inhibition reduces the size of lymph node metastases [65], highlighting the important role of lipid metabolism in metastatic potential. Glioma, the most common form of primary brain tumor in adult, was thought to strictly rely on glucose oxidation for energy production but this dependency has been recently re-investigated. Consistent with high expression of fatty acid oxidation enzymes in glioma, proliferation of human glioma primary cells mainly relies on OXPHOS metabolism through fatty acid oxidation [66,67]. A metabolic intra-tumor heterogeneity within glioblastoma has also been recently reported, in which aerobic glycolysis and OXPHOS cells co-exist and are associated with fast or slow-cycling cell capacity, respectively [68,69]. Finally, lipid metabolism plasticity has been recently emphasized by S.M. Fendt’s lab showing that lung and liver cancer cells can exploit an alternative fatty acid desaturation pathway to generate an unusual monosaturated fatty acid, called sapienate (cis-6-C16:1), and thus support cancer cell proliferation [70]. Taken together, all these recent findings uncover the important role played by fatty acid metabolism in cancer cells.

#### 2.1.3. Lactate as a TCA Carbon Source

Increased glucose consumption and lactate production are hallmarks of cancer cells [2]. Initially considered as a waste product of the Warburg effect (Figure 1), lactate is nowadays also recognized as an energetic fuel both in malignant and physiological conditions [13]. Even if still controversial, it has recently been hypothesized that lactate, secreted by glycolytic cells, can be oxidized by neighboring cells. Using high-resolution metabolomic technologies and stable isotope labelling in cancer cell lines (HeLa and H460), exogenous lactate was shown to increase lipid synthesis and mitochondrial respiration through its oxidation in mitochondria, and not by the cytoplasmic LDH (Figure 1) [71]. In line with these observations, DeBerardinis’ laboratory demonstrated that non-small-cell lung cancers (NSCLC) exhibit heterogeneous ^13^C-glucose oxidation, which can be influenced by their environment [72]. While all tumors display high glucose uptake and glycolysis, metabolic heterogeneity was identified within tumors. Indeed, intra-operative ^13^C-glucose infusions in patients showed oxidation of multiple nutrients in well-perfused tumor areas, including lactate as a potential carbon source for TCA cycle [72]. Lactate use in TCA cycle in NSCLC patients was next demonstrated by ^18^fluorodeoxyglucose uptake in patients with the most aggressive NSCLC, and ^13^C-lactate incorporation in TCA cycle metabolites [14]. Lactate is abundant in circulation, ranging from 1 to 2 mM, while pyruvate is 10 times less abundant. By comparing glucose and lactate incorporation in vivo, and categorizing tumors as lactate consumers depending on the lactate/3-phosphoglycerate (3PG, an upstream glycolytic intermediate) ratio, DeBerardinis’ lab observed that lactate contributes more to TCA cycle than glucose in NSCLC [14]. This contribution was also validated in another study in both lung and pancreatic xenograft mouse models, where the contribution of circulating lactate to TCA cycle intermediates exceeds that of glucose by about two-fold, but with a preference for glutamine, making a larger contribution than lactate in pancreatic cancer [13]. Finally, lactate-consumer tumors are more likely to progress, even if the ratio was measured at the time of resection of the primary tumor, in some cases, years before recurrence or metastases are detected.

### 2.2. Mechanisms Involved in Metabolic Switch from OXPHOS to Aerobic Glycolysis

Several reviews have already discussed the molecular or environmental mechanisms involved in cancer metabolic reprogramming [8,73]. In this part, we will focus specifically on the mechanisms involved in glycolysis regulation among OXPHOS tumors.

#### 2.2.1. Hypoxia

Hypoxic regions arise in tumors through uncontrolled rapid proliferation of cancer cells that is often associated with lack of functional vasculature and appearance of necrotic regions. As a consequence, nutrient and oxygen deprivation stimulates hypoxia-induced pathways. This outstanding research was recently recognized by attribution of the Nobel Prize in Physiology and Medicine to Pr. William G. Kaelin Jr, Sir Peter J. Ratcliffe and Pr. Gregg L. Semenza for describing “how cells sense and adapt to oxygen availability” [74,75,76]. They discovered the molecular mechanism by which a cell adapts to oxygen level variations, mechanism involving hypoxia-inducible factors (HIF) and HIF-prolyl-hydroxylases (PHD). Hypoxia is considered to be one of the main drivers of metabolic switch in tumor cells. Indeed, one of the most important changes induced by a decrease in oxygen concentration is an elevation of glycolytic flux associated with high glucose consumption. Some excellent reviews from J. Pouyssegur [77] and C. Simon [78] have recently been published. Metabolic switch toward glycolysis upon oxygen lack provides the benefit of being no longer dependent on aerobic respiration, therefore balancing oxygen consumption with oxygen supply. HIF-1α transcription factor enhances glycolysis by increasing transcription of glucose transporters (GLUT1-3) and glycolytic enzymes, including hexokinase 2 (HK2) and lactate dehydrogenase A (LDHA). HIF-1α also inhibits TCA cycle by up-regulating transcription of the pyruvate dehydrogenase kinase (PDK), which inactivates the pyruvate dehydrogenase (PDH), thereby preventing the conversion of pyruvate into acetyl coenzyme A (acetyl-CoA). HIF-1α is not only stabilized upon low oxygen concentration, but also under normoxic conditions by chronic oxidative stress [79] or oncogenic signals [80] through regulation of iron(II)/Fe^2+^ and 2-oxoglutarate, two other PHD enzyme co-factors in addition to oxygen.

Under hypoxia, anaerobic switch favors glycolysis and attenuates mitochondrial respiration. Although the mitochondrial NDUFA4L2 (NADH dehydrogenase ubiquinone 1 alpha subcomplex, 4-like 2) is an HIF-1α-target gene, hypoxia-induced NDUFA4L2 reduces oxygen consumption by inhibiting ETC Complex I activity and limiting Reactive Oxygen Species (ROS) production [81]. In addition, in mammalian cells, expression of the cytochrome c oxidase (COX) is oxygen-regulated leading to a switch between COX4-1 to COX4-2 isoforms, which optimizes the efficiency of respiration at different oxygen levels [82]. In this context, mitochondria are the primary sites of hypoxia-induced metabolic reprogramming involving HIF-1α-dependent transcription of mitochondrial pyruvate dehydrogenase kinase (PDK). In fact, PDK phosphorylates the pyruvate dehydrogenase complex (PDC) on three different sites [83]. PDK inhibits PDH, and further attenuates decarboxylation of pyruvate into acetyl-CoA. By doing so, an active PDK interrupts OXPHOS metabolism towards glycolysis. Recently, a phospho-proteomic approach identified a pool of phosphorylated AKT, which is translocated from cytoplasm to mitochondria during hypoxia [84]. This active AKT pool phosphorylates PDK1 on threonine 346 and inactivates PDC, which in turn converts metabolism from OXPHOS to glycolysis [84]. Finally, it has been shown in chemotherapy-resistant triple negative breast cancers that MYC and myeloid cell leukemia-1 (MCL1) genes are overexpressed, therefore promoting an OXPHOS metabolism [85]. Enhanced OXPHOS metabolism increases production of ROS, which in turn promote HIF-1α stabilization [79] and subsequently potentiate the enrichment of cancer stem cells [85]. Taken as a whole, these recent data suggest an increased level of complexity of metabolic heterogeneity within tumors, where the redox imbalance and HIF-dependent signaling pathway play important roles.

#### 2.2.2. Redox Metabolism, Role of BACH1

A new transcription factor, named BTB and CNC homology 1 (BACH1), has recently emerged as a central actor connecting redox balance to cancer metabolism reprogramming. BACH1, a member of cap ‘n’ collar (CNC) and basic region leucine zipper factor family, is able to bind to antioxidant response elements (ARE) and compete with the NFE2L2/NRF2 anti-oxidant factor [86]. In low oxidative stress conditions, BACH1 is stabilized due to low heme levels, prevents NFE2L2/NRF2 binding to ARE, and therefore acts as a transcriptional repressor of antioxidant genes. Interestingly, a new function of BACH1 has been identified in metabolic reprogramming in breast [87] and lung [88] cancer. Indeed, BACH1 was identified as a direct repressor of mitochondrial respiration through its binding to promoters of ETC-encoding genes [87]. Moreover, hexokinase 2 (HK2) and glyceraldehyde-3-phosphate dehydrogenase (GAPDH) have been identified among the strongest transcriptional targets of BACH1, suggesting that BACH1 stimulates glycolysis in lung cancer [88]. In addition, BACH1 is not only necessary but also sufficient to increase glycolysis in lung cancer cells and to promote their migration and invasion [88]. Similar to in lung cancer, BACH1 silencing in breast cancer cells promotes mitochondrial respiration and TCA cycle and concomitantly reduces glycolytic rate and lactate production [87]. Mechanistically, BACH1 regulates PDH activity by acting on PDK transcription [87]. Consistent with findings in breast and lung cancer, BACH1 expression has been shown to be inversely correlated with OXPHOS pathway in TCGA datasets from breast, pancreas, ovary, skin, lung, liver, colon and prostate cancer [87]. Taken together, these data indicate that BACH1 is a new common mechanism involved in OXPHOS repression and metabolic switch in cancer.

#### 2.2.3. Tumor Suppressor, PUMA

The tumor suppressor TP53 is one of the most common mutated genes in cancer, playing numerous roles in cell cycle, apoptosis, senescent and genome integrity maintenance [89]. TP53 is also involved in cancer metabolic regulation [90,91]. Indeed, TP53 prevents malignant progression by regulating metabolism at different levels, including by inhibiting aerobic glycolysis and triggering OXPHOS [92,93,94,95,96,97,98]. The majority of *TP53* mutations in human cancer are missense mutations that lead to the synthesis of mutant proteins often stabilized and accumulating at high levels in cancer cells [99]. In contrast to their wild-type (WT) counterpart, mutant TP53 promotes aerobic glycolysis in cancer cells, in part by enhancing glucose import through glucose transporter 1 (GLUT1) [100,101]. The mechanisms by which WT and mutant TP53 regulate the same metabolic pathways and their contributions to tumor progression are far from clear. In that context, a recent study has discovered a new paradoxical role for the WT form of TP53 in hepatocellular carcinoma, showing it plays a dominant metabolic role by promoting switch from OXPHOS metabolism to glycolysis through PUMA, the transcriptional target of TP53 [102,103]. By performing fluorescence resonance energy transfer assay and confocal immunofluorescence analysis, authors showed that PUMA suppresses the oligomerization of mitochondrial pyruvate carrier (MPC) leading to a decreased transport of pyruvate into mitochondria [102]. IKB kinase mediates phosphorylation of PUMA at serine S96 and S106, and is necessary to recruit PUMA from the cytoplasm to the mitochondria promoting its interaction with MPC, thus inhibiting pyruvate uptake [102]. Although the inactivation of MPC is already known to suppress OXPHOS metabolism, this study reveals a new metabolic role of PUMA in shifting metabolism from OXPHOS to glycolysis.

#### 2.2.4. Epigenetic Modifiers

Epigenetic modifiers are often altered or mutated in cancer and have been involved in tumorigenesis. In addition to the well-known nutrient sensors, such as AMP-activated protein kinase (AMPK) or mechanistic target of rapamycin (mTOR), metabolite abundance is also sensed by post-translational modifiers [104]. Indeed, these enzymes used metabolites as substrates, such as acetyl-CoA or acetyl donor, S-adenosylmethionine (SAM), O-linked Beta-N-acetylglucosamine (O-GlcNAc) [104]. The hexosamine biosynthetic pathway relies on glucose and glutamine uptake and is responsible for UDP-N-acetylglucosamine (UDP-GlcNAc) production. This end product is required for the synthesis of different extracellular glycopolymers (N- and O-glycans) and is also the substrate of O-GlcNAc transferase (OGT), providing O-GlcNAc post-translational modifications [105]. An elevated level of O-GlcNAcylation has been reported in various cancers and was shown to promote glycolytic program in breast cancer cells [106]. Mechanistically, high O-GlcNAcylation level stabilizes HIF-1α protein by diminishing α-KG levels [80], thereby promoting expression of GLUT1 [106]. In addition, the histone methyltransferase KMT2D is one of the most highly inactivated epigenetic modifiers in lung cancer, inactivation that confers a glycolytic vulnerability to tumors [107]. Indeed, loss of *Kmt2d* in Kras^G12D^ mice promotes lung tumorigenesis, and favors not only OXPHOS metabolism but also a glycolytic reprogramming through impairment of super-enhancers [107]. Mechanistically, *Kmt2d* loss impairs epigenomic signals of the circadian rhythm repressor *Per2* super-enhancer. This inhibits *Per2* expression, which in turn regulates multiple glycolytic genes. This study thus highlights a new regulation mechanism linking epigenetic modifier and circadian rhythm regulator to glycolytic reprogramming in lung cancer.

#### 2.2.5. Extracellular Matrix (ECM) Protein Remodeling

ECM remodeling within the TME, a well-known cell-extrinsic signal, has often been associated with concomitant elevated glycolysis. Christofk’s laboratory reported one of the first mechanisms linking together these processes [108]. By performing an unbiased analysis on breast cancer cell lines and tumors, they observed that the expression of hyaluronic acid (HA) receptor gene (*HMMR*) correlates with glycolytic metabolism [108]. Interestingly, by using a xenograft mouse model, authors reported an increased FDG uptake by tumors following hyaluronidase treatment, suggesting that ECM enzymatic digestion stimulates tumor glucose metabolism in vivo [108]. Mechanistically, they identified the zinc-finger protein 36 (ZFP36), which promotes the degradation of thioredoxin interacting protein (TXNIP) mRNA, and in turn decreases GLUT1 localization at the plasma membrane [108]. These data suggest that spatial heterogeneity in ECM composition may contribute to intra-tumor metabolic heterogeneity and modulation of tumor cell proliferation.

### 2.3. Consequences of Metabolism Heterogeneity in Tumors

In this part, we will discuss the consequences of the metabolic heterogeneity described above by focusing not on proliferation and tumor growth, which have been extensively reported previously, but on invasion and metastatic spread, as well as resistance to treatment.

#### 2.3.1. Invasion and Metastatic Spread

How cancer cell metabolic reprogramming impacts tumor cell invasion remains an important field of investigation. Although still highly debated, it was shown since a long time that metabolic stress, such as hypoxia or oxidative stress, promotes angiogenesis, invasiveness and metastatic spread. Indeed, by comparing melanoma cells from subcutaneous primary sites and visceral metastases, metabolomic analyses highlighted the key role of antioxidant defenses for counteracting ROS increase in metastatic nodules [55]. Consistent with these observations, mitochondrial mass and activity decline significantly in metastatic cells, with a preferential use of the serine/glycine pathway compared to primary sites [55]. Moreover, BACH1, which binds to ARE binding sites and competes with NFE2L2/NRF2, is required for metastasis in triple negative breast cancer [87] and in lung cancer [88,109]. Indeed, BACH1 induces a pro-metastatic transcriptional program by reducing OXPHOS metabolism. Many studies have shown that a metabolic switch is associated or necessary to promote the migration and invasion of tumor cells. For instance, highly invasive ovarian cancer cells exhibit a dramatic increase in oxygen respiration rate through glutamine anaplerosis [110]. It is likely that OXPHOS metabolism mediated by PGC-1α promotes breast cancer cell invasiveness and metastases [111]. Recently, in vivo evidence has shown differences in the metabolic and nutrient requirements involved in serine biosynthesis, necessary to activate growth signaling between the lung metastatic and the primary breast cancer sites [112]. A recent study using lung cancer cells revealed a metabolic heterogeneity within collective invasion packs of tumor cells [113]. Indeed, leader cancer cells -a specialized and highly invasive subtype localized at the tips of invading cell chains-use mitochondrial respiration, while follower cells -a poorly invasive trailing subtype- rely on high glucose uptake [113]. Lipids also contribute significantly to cancer metabolism by serving as a carbon source to fuel the TCA cycle and importantly by providing substrates for cell membrane generation. Indeed, fatty acid transporters, such as CD36 or fatty acid transporter protein (FATP), help tumor progression in oral squamous cell carcinoma, ovarian cancer and melanoma, by increasing fatty acid uptake [63,65,114]. A large metabolomic study performed on 17 melanoma PDX showed that the abundance of metabolites involved in protein methylation correlates with the metastatic cancer cell capacity [115].

Epithelial to mesenchymal transition (EMT) is involved in several processes related to cancer aggressiveness and invasion. Several transcription factors, such as SNAIL, TWIST, ZEB inducing EMT [116] also promote early steps of malignant transformation in particular in triple negative breast cancer [117]. In this study, the authors showed that ZEB1 expression promotes malignant transformation through a stemness-oxidative stress axis dependent of the methionine sulfoxide reductase MSRB3, while maintaining low levels of DNA damages and chromosomal instability [117]. Moreover, SNAIL was shown to inhibit mitochondrial respiration and to stimulate glycolysis [118]. Interestingly, EGFR-dependent activation of uridine 5-diphosphate (UDP)–glucose 6-dehydrogenase (UGDH) depletes UDP-glucose level, resulting in enhanced expression of SNAIL and increased lung cancer cell migration and metastasis in mice [119]. In line with these findings, decreased OXPHOS capacity is usually associated with EMT in many cancer types [120]. PGC-1α, the master regulator of OXPHOS metabolism, displays a lower activity in metastasis specimens than in primary prostate tumors. Furthermore, PGC-1α blocks EMT and metastasis [121]. These data suggest that modulation of the metabolic program, through PGC-1α, may influence prostate cancer metastasis and progression. Still, no difference in lipid peroxidation was detected in vivo in prostate metastases compared to primary tumors [121]. Loss of attachment from ECM is associated with increased ROS production and decreased glucose-derived pentose phosphate pathway [122]. Interestingly, enforcing an EMT-like phenotype induced by spheroid formation with lung cancer cells reduces lipogenesis and increases OXPHOS metabolism, independently of hypoxia [123]. On the basis of isotope tracing experiments, the authors showed that isocitrate and citrate are used for OXPHOS metabolism and NAPDH production to counteract ROS production [123]. In addition, silencing fatty-acid synthase enzyme induces EMT and enhances metastasis [123]. Taken as a whole, these data demonstrate the strong impact of metabolic regulation in EMT and metastatic spread.

#### 2.3.2. Resistance to Treatment

How metabolism reprogramming impacts response to treatment in cancer is an important question that remains unclear, and depending on cancer type, drugs used and anti-oxidant response. Diffuse large B lymphoma is a genetically and metabolically heterogeneous disorder [4,124]. However, patients are treated with the same chemotherapeutic treatment, combining cyclophosphamide, hydroxydaunorubicin, oncovin, and prednisone in association with immunotherapy anti-CD20 (R-CHOP), and still 40% of patients are resistant [125]. Interestingly, R-CHOP resistant tumors, characterized by low GAPDH levels, exhibit an OXPHOS metabolism, with a preference for glutamine carbon source, instead of glucose [125]. In contrast, high-OXPHOS ovarian cancers exhibit a better response to chemotherapy (based on carboplatin and paclitaxel regimen) than low-OXPHOS tumors, independently of the BRCAness status [38]. Interestingly, high-OXPHOS ovarian cancer cells are characterized by an elevated ROS content, while low-OXPHOS tumors exhibit elevated levels of glutathione intermediates [38]. An interesting question is to characterize the metabolism of chemoresistant cells. Despite an initial response to the standard cytarabine plus anthracycline-based chemotherapy, almost all acute myeloid leukemia (AML) patients relapse due to the existence of rare chemoresistant stem cell populations. By using PDX models, it has been shown that upon treatment, resistant cells increase oxygen consumption rate and exhibit elevated levels of ROS [126]. These different sets of data demonstrate the key role of metabolism and oxidative stress in response to chemotherapy in different cancer types.

## 3. Cancer-Associated Fibroblasts (CAF) Identity, Function and Metabolic Features

The role of fibroblasts and the concept of “seed and soil” have been considered as key actors in cancer since the 19th century [127]. Since then, their study has remained in constant progress. Although the precise origin of CAF is still unclear [128,129,130], defining CAF identity and precisely characterizing their functions are research topics that are in full expansion. Here, we will review a comprehensive knowledge of the different CAF subpopulations, functions and metabolic phenotypes in solid tumors.

### 3.1. Different CAF Subpopulations in Cancer: Distinct Identities and Functions

#### 3.1.1. Different CAF Identities

CAF represent one of the most abundant components of tumor micro-environment that provide a mechanical support to cancer cells but also control their proliferation, survival, metastasis and resistance to therapies [131]. CAF heterogeneity has recently been highlighted in several studies by combining the use of several markers and/or by developing new cutting-edge technologies such as single cell approach. Indeed, several markers, which are not -or only at low levels- detected in normal fibroblasts, were identified to characterize CAF, including fibroblast activated protein (FAP), smooth-muscle actin (SMA), fibroblast specific protein-1 (FSP1/S100A4), Integrin β1 (CD29), platelet derived growth factor receptor α or β (PDGFRα/β) and podoplanin (PDPN) [132]. Importantly, these different markers are not expressed similarly or simultaneously in CAF, thereby highlighting a strong degree of heterogeneity of CAF in cancer. By integrating the concomitant analysis of several CAF markers, four different CAF subpopulations (referred to as CAF-S1 to CAF-S4) have been discovered in breast and ovarian cancer [133,134]. CAF-S1 (FAP^High^ CD29^Med^ SMA^Med-High^ FSP1^Med^ PDGFRβ^Med-High^) and CAF-S4 (FAP^Neg-Low^ CD29^High^ SMA^High^ FSP1^Low-Med^ PDGFRβ^Low-Med^) express high levels of SMA and can be defined as myofibroblasts, while CAF-S2 (FAP^Neg^ CD29^Low^ SMA^Neg^ FSP1^Neg-Low^ PDGFRβ^Neg^) and CAF-S3 (FAP^Neg^ CD29^Med^ α-SMA^Neg^ FSP1^Med-high^ PDGFRβ^Med^) do not. CAF-S1 fibroblasts are defined by ECM and wound-healing gene signatures, while CAF-S4 are characterized by a perivascular contractile gene signature [133,134,135,136]. Several studies from bulk or single cell data from human cancer and mouse models confirmed the existence these different populations of ECM-enriched (CAF-S1) and contractile (CAF-S4) sub-populations [137,138,139,140,141,142]. Among the FAP^High^ CAF (CAF-S1) subpopulation, two distinct subsets exhibiting either a matrix-producing myofibroblastic phenotype (called myCAF) or an inflammatory CAF (named iCAF) were recently identified in different types of cancers [137,142,143,144,145,146]. Finally, the large number of FAP^High^ CAF-S1 fibroblasts recently sequenced at single cell levels in [142] reached a very high resolution of this subpopulation and identified eight different FAP^High^ CAF clusters, with three clusters belonging to the iCAF subgroup and five clusters to the myCAF subgroup [142]. Thus, iCAF and myCAF subsets can be—by themselves—subdivided into different clusters defined by specific processes and signatures, such as detox-iCAF, IL-iCAF, IFNγ-iCAF and ecm-myCAF, TGFβ-myCAF, wound-myCAF and IFNαβ-myCAF, respectively. Taken together, these findings highlight the existence of various CAF subpopulations in solid tumors and underline their relevance in various cancer types and across species.

#### 3.1.2. Different Functions of CAF Subsets in Metastatic Spread and Immunosuppression

While normal fibroblasts suppress tumor formation, CAF usually exhibit tumor-promoting activities, enhancing cancer cell proliferation, invasion, angiogenesis, inflammation and extracellular matrix (ECM) remodeling. Still, some CAF were shown to prevent cancer cell invasion [147]. These observations suggest that different CAF subsets in tumors can exert opposite roles. In that context, both CAF-S1 and CAF-S4 subsets accumulate in aggressive breast cancer (HER2 and TN) [133] and in metastatic lymph nodes [135]. Moreover, accumulation of the CAF-S1 subset in early luminal breast cancer is associated with late distant relapse [136]. Both CAF-S1 and CAF-S4 promote metastases through complementary mechanisms [135], confirming that high myofibroblast content is associated with poor prognosis in breast cancer [148]. Importantly, during the past years, some specific CAF subsets have also been shown to modulate anti-tumor immune response. Indeed, CAF immunomodulatory functions can affect both innate and adaptive immunity. CAF-mediated immunosuppressive function can be direct via the secretion of chemokines, such as CXCL12 or IL6, that can retain suppressive immune cells, or indirect through ECM remodeling, forming a physical barrier for immune cell infiltration [149,150,151,152,153]. In addition, FAP^High^ CAF-S1 fibroblasts are able to increase the content in regulatory T lymphocytes and inhibit the activity of effector and cytotoxic immune cells through a multistep mechanism. Indeed, FAP^High^ CAF-S1 are able to attract, retain, increase the survival of CD4+ CD25+ T lymphocytes and promote their differentiation into regulatory T cells (Treg) [133,134,154]. In line with the identification of several subsets among FAP^High^ CAF-S1, recent data has demonstrated that only specific ones, characterized by ECM accumulation, wound-healing signature and TGFβ-signaling, are associated with an immunosuppressive tumor environment [139,141,142,155]. In particular, ecm-myCAF and TGFβ-myCAF cellular clusters are correlated with the content in FOXP3^+^ T lymphocytes in breast cancer, and are able to increase PD-1 and CTLA-4 protein levels at the surface of FOXP3^high^ Tregs [142]. Interestingly, accumulation of ecm-myCAF, wound-myCAF and TGFβ-myCAF clusters is correlated with resistance to immunotherapy in melanoma and non-small cell lung cancer patients [142]. Moreover, TGFβ inhibition has been shown to reduce myofibroblast features, while increasing immunomodulatories properties in murine carcinomas, thereby providing the rationale of combining TGFβ and PD-1/PD-L1 in clinical settings [155]. All these data shed light on CAF heterogeneity within tumors, in particular on the FAP^High^ CAF-S1 subset, and demonstrate their different functions in tumor progression, metastatic spread, immunosuppression and resistance to immunotherapies.

### 3.2. CAF Metabolism

One of the first studies that evaluated CAF metabolism in cancer considered a symbiotic relationship between tumor cells and CAF [156], although this postulate is still debated. By performing proteomic and transcriptomic analyses from Cav-1-deficient stromal cells, the authors observed an upregulation of both myofibroblast markers and glycolytic enzymes (LDHA and PKM2), effect associated with tumor recurrence and poor prognosis in breast cancer patients [156]. Based on these observations, the authors postulated that cancer cells stimulate aerobic glycolysis in neighboring CAF. Glycolytic CAF then undergo myofibroblast differentiation and secrete high levels of pyruvate and lactate, which are in turn used by cancer cells as new energy fuels. This new concept was referred to as the “reverse Warburg effect”, and represents metabolic vulnerability that might be targeted therapeutically [156,157,158].

#### 3.2.1. Aerobic Glycolysis

In one of the first comprehensive analyses comparing metabolic variations between quiescent and proliferating fibroblasts, the authors used isotope labelling analyses [159]. This study revealed that quiescent primary dermal fibroblasts use glycolysis for baseline homoeostasis, although proliferating fibroblasts increased two-fold the glycolysis rate to supply with biomass requirements [159]. Interestingly, proliferating fibroblasts use glucose carbons through the pentose phosphate pathway to generate ribose-5-phosphate, while quiescent fibroblasts use those carbons to generate NAPDH to maintain redox homeostasis [159]. Human CAF from breast cancer, colon carcinoma and melanoma also exhibit a Warburg effect characterized by an increased glucose uptake and lactate production, while oxygen consumption is decreased [156,160,161]. Similarly, CAF from PDAC also predominantly use glucose carbons for aerobic glycolysis [162]. Taken together, these studies suggest that CAF preferentially oxidize glucose carbons to produce lactate.

#### 3.2.2. OXPHOS Metabolism

Metabolomic comparison between quiescent and proliferative fibroblasts demonstrated that only proliferating fibroblasts are able to use glutamine, in an anaplerotic process to replenish TCA cycle intermediates [159]. Similarly, glutamine is also used by CAF in PDAC to replenish TCA cycle. Consistent with these findings, CAF are sensitive to glutamine deprivation, while they are resistant to glucose starvation [162], suggesting the importance of this energetic preference. In ovarian carcinoma, transcriptomic analyses of two different CAF SMA^+^ subpopulations detected within the same tumor type, CAF-S1 (FAP^High^ CD29^Med^ SMA^Med-High^) versus CAF-S4 (FAP^Neg-Low^ CD29^High^ SMA^High^), highlighted for the first time a heterogeneous metabolic program among CAF [134]. Indeed, authors found that CAF-S4 exhibit an increased expression of genes related to ETC complexes, when compared to CAF-S1 [134], suggesting that CAF-S4 may be dependent on OXPHOS metabolism in contrast to CAF-S1. Importantly, this OXPHOS heterogeneity was also validated in head and neck cancer by RNA sequencing on single fibroblast cells [163]. Indeed, as in breast and ovarian cancers, authors were also able to distinguish two CAF subpopulations, FAP^High^ and MCAM^High^, similar to CAF-S1 and CAF-S4, respectively. They reported that FAP^High^ CAF exhibit enhanced glycolysis, as well as enriched arachidonic and linoleic acid metabolism, while MCAM^High^ rely on OXPHOS and TCA cycle [163]. Interestingly, it has been shown that ovarian cancer cells stimulate glutamine production in adjacent fibroblasts to satisfy their need [164]. All these findings suggest that different CAF subpopulations are not only associated with various functions, but also with distinct metabolic programs, which rely either on glycolysis or mitochondrial respiration, as observed in malignant epithelial tumor cells.

### 3.3. Metabolic Crosstalk between Cancer Cells and CAF 

In this part, we will focus on the crosstalk between tumor cells and their surrounding CAF. How does CAF influence tumor cell metabolic state? Reciprocally, how cancer cells impact the metabolism of their surrounding CAF? Here, we will discuss the different metabolic interactions between tumor cells and CAF that promote tumor growth and progression (Figure 2).

#### 3.3.1. CAF Cooperation to Meet the Energetic Demands of Proliferative Tumor Cells

To understand the crosstalk between tumor cells and CAF, the use of PDX mouse models has been useful, as PDX make it possible to differentiate tumor cells from human origin, and stromal cells from mouse origin. Interestingly, it has been reported that ovarian cancer cells in PDX keep the metabolic features harbored by the original tumor cells in patients [38]. These findings suggest, at least in PDX models, that tumor cells have their own metabolic identity and that they might be able to educate the stromal compartment to fit their energetic demands. In line with these assumptions, metabolic profiling of both stromal and cancer cells in colorectal cancer PDX are stable and remain comparable to the patient ones [165]. The concept of metabolic cooperation between tumor cells and CAF was initially evoked by the “reverse Warburg effect” [156,157,158]. Exchange of amino acids between cancer cells and CAF has recently been reported to help cancer cells in their high energetic requirements [166,167,168]. In PDAC, stroma-associated pancreatic stellate cells are essential to sustain cancer cell metabolism through secretion of non-essential amino acids [45]. Metabolomic analyses revealed that alanine was highly secreted by stellate cells and act as an alternative carbon source to support OXPHOS metabolism in tumor cells through transamination (Figure 2) [45]. Mechanistically, autophagy promotes alanine release by stellate cells that leads to increased PDAC proliferation. Interestingly, a same kind of metabolic cooperation also exists in ovarian cancer [164]. Indeed, ovarian cancer cells act on their surrounding fibroblasts to satisfy their demand of glutamine carbons to replenish the TCA cycle and to support cancer growth by increasing purine and pyrimidine biosynthesis pathways (Figure 2) [164]. Reciprocally, authors highlighted that cancer cells provide lactate and glutamate to CAF, thereby enhancing the TCA cycle and allowing glutamine production by CAF (Figure 2) [164]. Interestingly, YAP/TAZ-dependent pathway and ECM stiffening also exert metabolic reprogramming to sustain cancer cell proliferation (Figure 2) [167]. Indeed, stiffness promotes both glycolysis and mitochondrial respiration in CAF, whereas only glycolysis rate is increased in cancer cells. More precisely, matrix stiffening favors glutamine consumption and subsequent aspartate release by CAF; aspartate is next uptaken by cancer cells. This metabolic crosstalk sustains nucleotide biosynthesis and cancer cell proliferation, as well as redox homeostasis in CAF (Figure 2) [167]. Remarkably, these complementary metabolic exchanges have not only an impact on tumor cell features but also increase CAF contractility. ATF4 up-regulation induced by p62 deficiency in CAF—a key component of the regulation of mTOR pathway—activates glucose carbon flux to the TCA cycle through a pyruvate carboxylase-asparagine synthase route that leads to asparagine production as a source of nitrogen for both stroma and tumor epithelial proliferation [169]. Despite these important findings, many questions remain: How does metabolic interactions occur between CAF and cancer cells? Which CAF subpopulation(s) could be implicated in this reciprocal crosstalk? A better understanding of these remaining issues would provide important information for developing therapeutic strategies to target both cancer cells and CAF in the same tumor ecosystem.

#### 3.3.2. Modulating the Redox Homeostasis

There is an amount of evidence supporting the relationship between metabolic reprogramming and redox balance, both in tumor cells and their microenvironment [6]. CAF metabolic reprogramming has been highlighted in PDAC [170]. p62 silencing in stromal cells impairs metabolic detoxification—in a NRF2- and NF-κB-independent manner—and increases IL-6 and TGFβ secretion (Figure 2) [170]. By using stable isotope tracing, authors reported a concomitant reduction of glutamine consumption and glycolytic flux through the pentose phosphate and serine pathways, leading to a decreased GSH synthesis and NADPH/NADP+ ratio (Figure 2) [170]. In contrast, in squamous cell carcinomas, metabolic exchanges between cancer cells and CAF promote glutathione synthesis through glutamate consumption, and maintain redox balance in CAF [167]. These results reflect the complex relationship between metabolic reprogramming and redox homeostasis, highlighting the importance of a deeper characterization of CAF subpopulations involved in these multiple processes.

#### 3.3.3. Paracrine Fatty Acid Secretion

In PDAC, an unanticipated lipid crosstalk between CAF and cancer cells has recently been described [166]. Pancreatic stellate cells produce an oncogenic lysophosphatidic acid (LPA) signaling that promotes growth of tumor cells (Figure 2) [166]. Moreover, during tumorigenesis, pancreatic stellate cells activation into CAF undergoes a lipid metabolic shift characterized by a loss of lipid storage [171]. By using palmitate and oleate isotope tracing experiments, a fatty acid metabolic cooperation between PDAC cells and pancreatic stellate cells has been uncovered. These interactions promote the accumulation of lysophospholipids and triglycerides in PDAC cancer cells, coming from stroma-derived fatty acids. In addition, they lead to the secretion of autotaxin, which gives rise to LPA that further supports membrane synthesis to sustain PDAC cell proliferation and migration [166]. Still, authors do not investigate differences between iCAF and myCAF, while they show a different spatial distribution in TME in PDAC [143]. In this sense, several classes of lipids have been already reported to promote metastasis in different tumor types, such as oral carcinomas [65] and melanomas [114] by modulating natural killer cells and other immunosuppressive populations [172]. Taken as a whole, these data demonstrate that metabolic crosstalk between cancer cells and CAF provide multiple pro-tumorigenic functions that deserve further investigation.

## 4. Conclusions

In conclusion, a tumor is now considered an ecosystem where all different types of cells, malignant and non-malignant, interact with each other, in a cooperative but also a competitive way. In this review, we discuss metabolic heterogeneity and requirements of both tumor cells and CAF. We also address how these two populations metabolically interact with each other. Targeting the metabolic crosstalk between tumor cells and CAF may represent a potential therapeutic avenue in the cancer field.

## Figures and Tables

**Figure 1 cancers-13-00399-f001:**
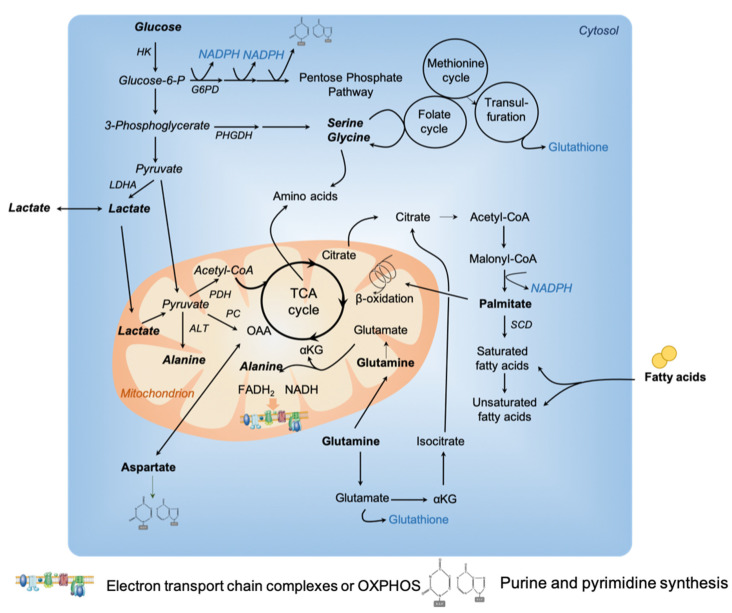
Carbon-source preferences and use in cancer metabolism. Glucose carbons are oxidized to produce pyruvate trough glycolysis. Glucose carbons fuel the tricarboxylic acid (TCA) cycle and subsequent oxidative phosphorylation (OXPHOS) in mitochondria. Intermediates of glucose metabolism are diverted from glycolysis and used for biosynthetic purposes such as the pentose phosphate pathway, the serine pathway, and the one carbon metabolism–Folate–Methionine cycle and the transulfuration pathway. The TCA cycle can be fueled through anaplerotic reactions involving glutamine, alanine, aspartate, serine, palmitate (or other fatty acids, not represented) and lactate. Carbon sources are highlighted in bold, anti-oxidant in blue and enzymes in italic. αKG: alpha-ketoglutarate; ALT: Alanine aminotransferase; CoA: Coenzyme A; FAD: flavin adenine dinucleotide; G6PD: Glucose-6-phosphate dehydrogenase; HK: Hexokinase; LDHA: Lactate dehydrogenase A; NADPH: Nicotinamide adenine dinucleotide phosphate; OAO: Oxaloacetate; PC: Pyruvate Carboxylase; PDH: Pyruvate dehydrogenase; PHGDH: Phosphoglycerate dehydrogenase; SCD: stearoyl-CoA desaturase.

**Figure 2 cancers-13-00399-f002:**
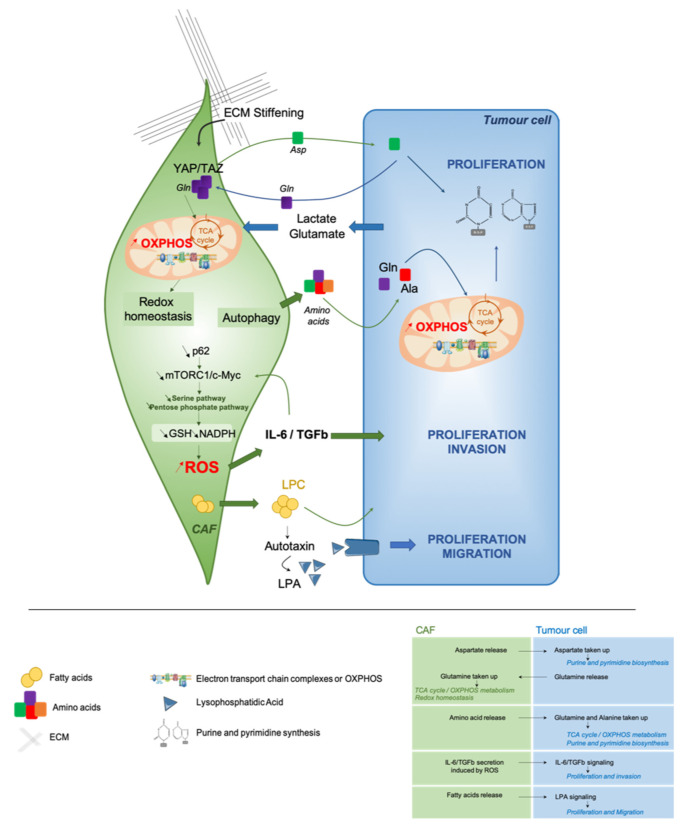
Metabolic crosstalk between CAF and tumor cells. CAF have the ability to support cancer cell growth, proliferation and migration/invasion by different metabolic mechanisms involving extra cellular matrix (ECM) stiffening, redox homeostasis, autophagy and lipid secretion. ECM stiffening, through YAP/TAZ signaling promotes glutamine taken up by CAF to fuel the TCA cycle and to maintain redox homeostasis and aspartate release. Autophagy results in the release of amino acids, such as glutamine (Gln) and alanine (Ala), which can be taken up and used by cancer cells to replenish the TCA cycle and OXPHOS metabolism. Decrease mTORC1/cMYC signaling in CAF reduces serine and pentose phosphate pathways leading to a redox imbalance. Therefore, increasing reactive oxygen species (ROS) accumulation and the secretion of IL-6 and TGFβ in the microenvironment. This secretion promotes cancer cell proliferation and invasion. CAF release lipids, including lysophosphatidylcholines (LPC) into the microenvironment. LPC can be converted to lysophosphatidic acid (LPA) by extracellular autotaxin. LPA induces cancer cell proliferation and migration through the LPA receptors on cancer cells. The arrows represent the metabolites secreted or released by CAF (green) or tumor cells (blue). GSH: Reduced glutathione.

## Data Availability

Not applicable for a review.

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
