# Peer review of "Tumor Cells and Cancer-Associated Fibroblasts: An Updated Metabolic Perspective"

_cancers, 2021, doi:10.3390/cancers13030399_

Round 1
Reviewer 1 Report
This review article is really well prepared and comprehensive. All aspects on the tumor cell and CAF-tumor cell interaction have been nicely illustrated and recent literature data have been included.
I think this paper represents a very nice piece as literature as it is.
Author Response
We are very grateful to the Reviewer for her/his positive evaluation of our work. We thank him/her for the careful reading of our manuscript.
Reviewer 2 Report
Authors: Gentric et al., 2021
- Is the topic important?
The authors review an important and emergent topic in cancer research on tumor metabolics and the interplay between cancer cells and the TME in this regard.
- Is it covered effectively?
This is a very well written, detailed, yet approachable review. The authors follow a systematic approach in which they discuss Metabolic heterogeneity, amino acids and their role in augmenting/modulating cancer cell metabolics and highlight the importance of alanine, glutamine, and serine, etc. They then discuss the oncologic impact of the metabolic pathways modulated by cancer cells through discussion of invasiveness, therapy resistance. Finally, the authors delve into the complex interplay between CAFs and cancer cells with respect to tumor metabolism and discuss in detail the reverse Warburg effect as well as other key examples of crosstalk between tumor cells and CAFs.
- Do the authors provide cogent synthesis of the information?
Yes. This is a detailed yet easily approachable review. The information is presented in a clear manner from the biochemical pathways that are altered to direct clinical implications on prognosis of alterations in metabolic pathways in multiple cancers. The authors also identify a possible biomarker and therapeutic target in CD36 as a fatty acid transporter which is associated with poor prognosis and heightened metastatic potential in several cancers. They authors highlight that abrogation of CD36 resulted in reduced LN metastasis size and reduced tumor growth in prostate cancer PDX models. The authors also emphasize the possible paradigm shift in the understanding of the role of lactate and pyruvate production as part of the Warburg effect and demonstrating that these products may have profound roles in crosstalk with CAFs as well as a crucial carbon source in the TCA cycle.
- Are appropriate references included?
Yes.
- Is it well written?
Yes. This is a well-organized elegantly written review that is detailed and approachable for a wide audience including basic science researchers, clinicians, and students. There are just a few very small wording changes that could be made including:
- Line 271: instead of stating “we would like to focus”, would be better so state “ we will focus…”
- Line 409. No need for “investigations”. It can simply be “important field of investigation”
- Lines 109-110. The citations and the number reference appears convoluted. Would perhaps be better to simply provide the numbered references in standard fashion.
- Are they missing anything?
- no
- Are the figures useful?
- The figures are detailed and the legends are informative. A small comment regarding Figure 2, which is complex. A small table on either side of the figure with end products and their final destination/role in augmenting the cancer cell metabolism would be helpful. For example, a table for lactate could demonstrate: “back to TCA cycle”, “taken in by CAFs in exchange for amino acids”, etc. as a summative statement on the figure as a whole.
Author Response
Points 1-3: We are very grateful to the reviewer for his/her positive and constructive evaluation of our work. We would also like to thank the reviewer for his/her comments and advices.
Point 5: The required modifications (in apparent into the new text) have been added in this new version of the manuscript, as requested: # 2.2, line 260, “we will focus”; # 2.3.1, line 386-7, “important field of investigation”; and references corrected into the new text, # 2.1.1, p4, line 114. We thank the reviewer for his/her careful reading and corrections.
Point 7: We thank the reviewer for this proposition. As requested, we have tried to provide a new Figure 2, which could be clearer and detailed, by adding a small table, as advised by the reviewer. In addition, we added new indications in the Figure legend to help in the understanding: “The arrows represent the metabolites secreted or released by CAF (green) or tumor cells (blue)”.
Reviewer 3 Report
The topic of the key role of cancer metabolic reprogramming has been well discussed in literature and although the Authors try to treat the subject from a new perspective not always is highlighted. The topic could be interesting and useful to the scientific community but more appropriate and recent literature works have to be included. The figures offers a succinct and clear schematic representation of the topic. Therefore the study is recommended for publication but after corrections.
- Page 2 Line 54- Figure ref should be moved into the paragraph.
- Page 2 Lines 69_71- “In line with these observations, a new way of understanding metabolic reprogramming in the cancer field has emerged: the metabolism of tumor cells is no more only compared with their normal counterparts, but also between tumors from the same cancer type.” Reference should be included.
- About paragraph “2.1. Metabolic heterogeneity and carbon-source preferences” The Authors should include the role of glutamine as carbon source (including references such as: Metallo et al. 2011; Davidson et al., 2016, Gaglio et al., 2020).
- Page 3 Line 96- The Authors should add references of research articles.
- Page 4 Line 107- Alternative TCA cycle branch sustained by glutamine reductive carboxylation should be added.
- Page 4 Line 112 Reference Gaglio et al., 2016 should be included.
- Glutamine paragraph should include the link of glutamine in cancer invasiveness (Yang et al., 2014).
- Serine paragraph should include role of PHGDH in in vivo metastasis (Rinaldi et al., 2020).
Author Response
We thank the reviewer for his/her comments and recommendations to improve our manuscript.
Point 1: The figure reference has been added in the paragraph itself, as recommended.
Point 2: As requested, references have been added in the new version of the manuscript p2, #2.1, line 72 ([4, 5, 12]).
Point 3: We have now included these references [22, 32, 34] in the new version of the manuscript p4, #2.1.1 Glutamine, lines 107 to 114.
Point 4: This opening paragraph (p3, #2.1.1 Amino acids, lines 95-100) is just a rapid introduction on the role of amino acids as new carbon source. As requested, references of recent research articles can be found in the following paragraphs.
Point 5: We have now added the notion that alternative TCA cycle sustained by glutamine reductive carboxylation in the new version of the manuscript (p4. lines 107-110). We thank the Reviewer for this recommendation.
Point 6: As recommended, this reference has been included in the new text, as reference [32].
Point 7: We have discussed about the role of cancer cell metabolism on cancer migration and invasion within a dedicated paragraph, with the most recent corresponding references herein, in particular this one recommended by the Reviewer (#2.3.1 Invasion and metastatic spread, p9, Reference [110]).
Point 8: As recommended, we have now added this study from Sarah-Maria Fendt’s lab in the new version of the text: #2.3.1 Invasion and metastatic spread, p10, Reference [112]. We thank the Reviewer for this recommendation.
Reviewer 4 Report
The review focuses on metabolism in cancer. Most of the review corresponds to general considerations on metabolism in cancer and only the last part deals with CAF metabolism. The first part of the review describes the different sources of carbon, including amino acids, fatty acids, lactate. A second part is focused on the mechanisms of switch from OXPHOS to aerobic glycolysis. Then the case of CAFs is discussed with a first part on their heterogeneity and a second one on their metabolism. If the review is of good quality, I have several recommendations to improve it.
Major points:
- The title of the review is not adequate as most of it deals with general considerations on metabolism in cancer and only the last part is focused on CAF metabolism. I would rather propose something like "metabolism and cancer: an update view".
- In many places, what is described is the combination of what has been observed in cancer cells and also in CAFs, which makes it difficult to summarize what is common and what is different. It would be interesting to have a section of this review describing specifically the differences between cancer cells and CAFs in terms of metabolism.
- Metabolism in immune infiltrating cells is not considered, it would be interesting to have a part on this.
- It is difficult to determine if each type of cancer has its own metabolism or if there are common features, which are always conserved among all types of cancers. The authors should discuss this point.
Minor points:
- The authors should refer to the work of Tong et al, 2020, BBA Rev Cancer concerning CAF metabolic heterogeneity
- One piece of work lacking in this paper is the acquisition of CAFs properties by mesenchymal stem cells and the metabolism of MSCs in cancer situation. This should be discussed and at least a few papers such as Quante, 2011, Cancer Cell/ Lazennec et Lam, 2016 BBA Rev Cancer / Misrah 2008 Cancer Res/ cited.
- The work of Grauel et al., 2020, Nat com on CAF heterogeneity is not discussed
Author Response
We are grateful to the Reviewer for her/his positive evaluation of our work, and the high rating given by the Reviewer in the different aspects of our manuscript.
We would like to emphasize that our review addresses the most recent findings of metabolism considering both cancer cells and CAF, as mentioned in the title. We are afraid that referring only to cancer cells in the title could be a bit misleading to the readers. Still, as recommended, we now mentioned in the title that our review is an updated version of this metabolic view.
As mentioned all along the text, we have specified what is specifically referring to cancer cells or CAF. In particular, we discussed CAF metabolism in the different CAF subpopulations recently identified in cancer in the paragraph #3.2. This is new and strictly based on the most recent studies on CAF heterogeneity in cancer.
We agree with the Reviewer that immune cell metabolism is interesting and key in cancer development. But this is a real field of research by itself, with reviews specifically dedicated on this field. As indicated in our title, we aimed at focusing on the metabolism of cancer cells, CAF and their reciprocal interaction, as initially discussed with the Editors.
As recommended, we have now inserted the references in the new version of the Text (References #128, #129, #130 et #155. We thank the Reviewer for these recommendations.
As suggested by the Reviewer, we have now discussed in more details the work from Viviana Cremasco’s lab (Reference #155), #3.1.2, p12, lines 515-522). We thank the Reviewer for this interesting suggestion that improves the quality of our manuscript.
Round 2
Reviewer 3 Report
The revised version of manuscript can be accepted.